

# Effectiveness of Sentinel-1 and Sentinel-2 for Flood Detection Assessment in Europe

Angelica Tarpanelli[1], Alessandro C. Mondini[1], Stefania Camici[1]

[1]1Research Institute for Geo-Hydrological Protection, National Research Council, Via Madonna Alta 126, 06128 Perugia

*Correspondence to*: Angelica Tarpanelli (angelica.tarpanelli@irpi.cnr.it)

**Abstract.** Inundation is one of the major natural hazards in Europe. The evaluation of the flood hazard and risk is not straightforward mainly due to the monitoring system that is poor or not uniformly distributed in the territory. The ESA Earth Observation Program, including a series of satellites, Sentinels, for the operative observation of the natural phenomenon, e.g. the inundations, can potentially reduce the gap. Sentinel-1 (SAR) and Sentinel-2 (optical) are demonstrated suitable for

mapping flooded areas, but despite the medium-high spatial and temporal resolution of the sensors, the mapping of inundated territories is often partial or missing. The objective of this study is to evaluate through a synthetic study the effectiveness of the Sentinel-1 and Sentinel-2 in the systematic assessment of floods in Europe, where the flood events have duration ranging from some hours to a few days. To reach the target, we analysed ten years of river discharge data over almost 2000 sites in Europe and we extracted flood events over some established thresholds, as proxies of flood riverine inundations. Based on the

revisit time of the satellites constellations and cloud coverage, we derived the percentage of potential inundation events that Sentinel-1 and Sentinel-2 could be able to observe. Results show that assuming the configuration of a constellation of two satellites for each mission and considering the ascending and descending orbit, on average the 58 % of flood events are potentially observable by Sentinel-1 and only the 28 % by Sentinel-2 due to the cloud coverage.

**Short summary:** We analysed ten years of river discharge data over almost 2000 sites in Europe and we extracted flood events, as proxies of flood inundations, based on the overpasses of Sentinel-1 and Sentinel-2 satellites to derive the percentage of potential inundation events that they could be able to observe. Results show that on average the 58 % of flood events are potentially observable by Sentinel-1 and only the 28 % by Sentinel-2 due to the obstacle of cloud coverage.

## 1 Introduction

According to the European Directive 2007/60/EC (Directive 2007/60/EC), "flood" is defined as "the temporary covering by water of land not normally covered by water" (art. 2.1) and it includes "floods from rivers, mountain torrents, Mediterranean ephemeral water courses, and floods from the sea in coastal areas, and may exclude floods from sewerage systems" (art. 2.1). Floods are one of the most common and costly natural hazard (Jongman et al., 2012). Catastrophic floods endanger lives and

cause human tragedy as well as heavy economic losses. According to the data elaborated by the International Disaster Database



(EM-DAT, 2019; Luo et al., 2015) from 1980 and 2021 (July), in Europe 614 flood events produced more than 4 thousand fatalities, 12 Million of affected people and 141 Billion of US$ of total damage. After the dramatic floods occurred in 2002, the European administrative authorities agreed to draw up guidelines for flood prediction, prevention and mitigation described in the well-known Directive 2007/60/EC (2007) on the assessment and management of flood risk. The main target of the

Directive is to reduce the adverse consequences on human health, the environment, cultural heritage and economic activity associated with floods. For this purpose, it is required for each Member States to develop three kinds of products: i) a preliminary flood risk assessment, in order to evaluate the level of flood risk in each river basin and select areas that are particularly critical to flood risk, (ii) flood hazard and risk maps under specific scenarios of inundation and (iii) flood risk management plans.

For the evaluation of flood hazard, vulnerabilities and risk, the maps of historical flooded events are of paramount importance because they represent the benchmark for calibrating the parameters of the hydrodynamic models that provide precious information on the evolution of flood in the floodplain and the environment around. Furthermore rapid mapping of new events is fundamental for early warning activities and then mitigating its impact on society. The knowledge of the flood delineation helps stakeholder in the decision – making and in the territory planning and supports the local authorities in the activities aimed

to protect lives and properties.

Hydrodynamic models can provide a valuable tool for delineating the flooded area and many examples can be found in literature at the local, regional and global scale (Neal et al., 2012; Bates and De Roo, 2000; David et al., 2011; Yu and Duan, 2014; Pappenberger et al., 2005). Often the modelling is carried out before the flood event occurs in order to collect inundation scenarios and identify the edge of the potential flooded areas.

Useful support to flood monitoring, modelling and mapping is offered by satellite sensors launched for the Earth Observation (Di Baldassarre et al., 2009; Domeneghetti et al., 2019). Indeed, remotely sensed data can contribute to monitor emergency situations, supporting the development of an efficient early warning system, the acts of evacuations and the impact of future flooding. From the medium to high resolutions, optical sensors (e.g. MODIS, MERIS, Landsat, SPOT, IKONOS, GeoEye, WorldView) have been produced imageries used for monitoring inundation events pre- during and post-crisis (Sakamoto et

al., 2007; Ogilvie et al., 2015). During the rainfall events, however, due to the incapability of the optical sensors to penetrate clouds, the delineation of the boundaries of flooded areas is hampered in near real-time. They can be useful in the next phase of the flood extension and damages assessment (Uddin et al., 2019; Caballero et al., 2019).

Alternatively, the active microwaves of Synthetic Aperture Radar (SAR) sensor penetrate clouds and provides day and night images. The comparison between images pre-and post-event is used to map inundated floodplain (Uddin et al., 2019; Anusha

and Bharathi, 2019) or inundation levels (Cohen et al., 2018; Clement et al., 2018).

The availability of these two instruments, optical and SAR, in the satellite platform assurances a global coverage and, in most of the cases, a cost-free source of data.

On these bases, the ESA Sentinels constitute the first series of operational satellites responding to European operational and policy needs of the Global Monitoring for Environment and Security (GMES) program. They are planned to provide global





coverage of environmental parameters with high spatial and temporal resolutions (Berger et al., 2012; Aschbacher and Milagro-Pérez, 2012). Among the satellite constellations developed for natural resource management and climate/environmental research, Sentinel-1 and Sentinel-2 are the most suitable for mapping flooded areas (Malenovsky et al., 2012).

Starting from the availability of satellite sensors, the European Copernicus programme provides the Civil Protection services of the Member States with the Emergency Management Service, a rapid mapping service for the production of damage assessment maps caused by natural or man-made disasters. The service offers a 24h/7d contact point for receiving activation requests, previously authorised by the European Commission through the Emergency Response and Coordination Centre (ERCC) at DG ECHO, and takes care of the whole service chain up to the delivery of the finished map products through a public portal (http://emergency.copernicus.eu). The Copernicus Emergency Management Service has two components: i) mapping (rapid mapping and risk and recovery mapping), to provide digital and vector formats information based on satellite imagery for geospatial analysis to support decision-making by emergency managers; ii) Early Warning System, through the European Flood Awareness System (EFAS), to strengthen the preparedness of national and local authorities, and to support preparatory measures before major flood events strike.

Since the first lunch of Sentinel-1A in 2014, a series of flood events have been mapped in Europe thanks to Sentinel-1 and Sentinel-2 imageries. Examples from the Copernicus Emergency Management Service affect all the European Countries, from the past (11 January 2016 in Northern Ireland) to the more recent flood events (13-16 July 2021, in Germany, Belgium, Switzerland, Netherlands). Although the potential of Sentinel missions has been highlighted and well described in literature (Cao et al., 2019; Plank, 2014), and despite the use at operational level shows an effective functionality, still many flood events are not mapped due to the spatial and temporal limits of the satellite measurements. In fact, the spatial and temporal resolutions, although improved with respect to the past missions, can be not sufficient to map the maximum extension of the flood event when the evolution of a flood event covers a timespan from some hours to a few days.

On this context, the objective of this study is to evaluate the effectiveness of a systematic use of Sentinel-1 and Sentinel-2 in the detection of floods in Europe, through a frequency analysis of the actual passages of the satellites over the rivers.

To reach the target, we collected and analysed 10 years of river discharge data over almost 2000 sites in Europe and we extracted the flood events from each discharge hydrograph through the use of specific thresholds of river discharge, as proxies of flood riverine inundations. Based on the revisit time of S1 and S2 over Europe, we synthetically derived the percentage of inundation events potentially observable. The reliability of the results was tested through a real case analysis over three stations where the river discharges were available during the working period of Sentinel-1 and 2.

## 2 Materials and Methods

This section contains a brief description of the satellite data from optical and SAR sensors, the observed time series of river discharge for the Europe and the main procedure adopted for the analysis.



## 2.1 Sentinel-1 SAR imagery and their current use in flood detection/mapping

The Sentinel-1 mission is a C-band Synthetic Aperture Radar (SAR) constellation of two polar-orbiting satellites. Sentinel-1 comprises a C-band SAR sensor at 10 m of spatial resolution to provide a high revisit time (6 days in constellation mode) all-weather day-and-night supply of imagery (Torres et al., 2012). Sentinel-1 ensures the continuity of C-band SAR data, building on ESA's and Canada's heritage SAR systems on ERS-1, ERS-2, Envisat and Radarsat, providing wide swath coverage and frequent revisit in dual polarization. Being an active microwave sensor, SAR penetrates clouds and provides images in both day and night. Smooth water surface, which reflects the radar radiation away from the sensor, shows low backscatter values. The weak return signal is represented by dark tonality on radar images. Under windy conditions and/or the presence of vegetation, the water surface gets rough and the backscatter increases, reducing the contrast between flooded and non-flooded areas. In this situation the detection of flooded areas is more difficult.

The capability to estimate inundation events from Sentinel-1 is already demonstrated by several studies (Landuyt et al., 2019; Notti et al., 2018; Huang et al., 2018), also in Europe (Amitrano et al., 2018; Bioresita et al., 2018; Twele et al., 2016). An image during or after the event is used to map the inundated floodplains. The comparison with a pre-event imagery (change detection approach) allows to obtain information on the permanent water (Uddin et al., 2019; Anusha and Bharathi, 2019; Tarpanelli et al., 2013). When detecting changes in SAR images, the customary way of comparing a pair of multitemporal images is the application of the log-ratio operator (Bazi et al., 2005; Carincotte et al., 2006; Martinez and Le Toan, 2007; Celik, 2010; Takeuchi, et al., 1999), which is defined as the logarithm of the ratio of the backscattered signals. Methods for the classification of flooded areas in the measures of changes are based on region-growing (Schumann et al., 2011), statistical active contour model (Mason et al., 2007), composite image (Long and Trong, 2001), grey level dependence method (Seiler et al., 2009), fuzzy C-means (Amici et al., 2004), supervised classification (Townsend, 2002).

The flooded areas are also classified applying the largely used method of histogram (or radiometric) threshold (Yonghua et al., 2007; Matgen et al., 2007; Martinis et al., 2009; Mason et al., 2012; Giustarini et al., 2013; Hostache et al., 2009; Tarpanelli et al., 2013) that allows for the "flood" and "non-flood" pixels to be discriminated by means of a representative backscattering coefficient threshold value. Oberstandler et al. (1997) introduced the visual interpretation approach, stating that it is able to provide more accurate results than an automatic procedure and many studies were based on that (Matgen et al., 2007; Tarpanelli et al., 2013).

The high revisit time of the satellite is sufficient to detect flood inundations of a duration longer than one week, but often fails for flash floods. The revisit time of the satellite is 12 days for one satellite, but if we consider the constellation of 2 satellites (A and B) and the ascending/descending orbits, the revisit time at the European latitude increases at 2-3 days. Despite this resolution seems high, often it is not sufficient to detect the maximum spatial extension of a flood event.





## 2.2 Sentinel-2 optical imagery and their current use in flood detection/mapping

Sentinel-2 routinely delivers high-resolution optical images globally, providing enhanced continuity of SPOT (Satellite Pour l'Observation de la Terre)- and Landsat-type data. Sentinel-2 carries an optical payload with visible, near infrared and shortwave infrared sensors comprising 13 spectral bands at 10 m, 20 m and 60 m spatial resolution with a swath width of 290

130 km. Being an optical sensors, Sentinel-2 can observe floods only during the daytime and in good weather condition, because the solar light cannot penetrate the clouds in the visible range (Drusch et al., 2012). Under these conditions, the number of chances to detect the maximum extent of a flood event is reduced because the cloud coverage adds to the satellite's revisit time.

For optical data, water areas are generally detected by using near-infrared (NIR) band because in the NIR band, the reflectance

of water surface is in general much lower than the reflectance given by other land cover types. However, optical images can be easily affected by atmospheric conditions, surface reflection of sun light, and water turbidity making difficult to set a common threshold value for flood detection. To mitigate the problem, band algebra is used, e.g. Normalized Difference Vegetation Index (NDVI) and/or Normalized Difference Water Index (NDWI), computed from combinations of visible, NIR and Short-Wave Infrared (SWIR) bands (Takeuchi et al., 1999; Townsend et al., 1998; Seiler et al., 2009; Sakamoto et al.,

2007). Commonly, the flooded area detection is derived thresholding such indexes. Alternatively, numerous other techniques such as Intensity Hue and Saturation, IHS (Yonghua et al., 2007; Goffi et al., 2020), Principal Component Analysis, PCA (Gianinetto et al., 2006), Support Vector Machine, SVM and C-mean clustering (Longbotham et al., 2012) are employed to detect flooded areas from the optical imagery.

## 2.3 *In situ* data of river discharge

We used river discharge time series recorded in the GRDC dataset (The Global Runoff Data Centre, 56068 Koblenz, Germany) joint with Italian and Spanish Basin Authorities datasets collecting data from 1913 to 2016 (see Figure 1a) for a total of 1957 in-situ gauged stations across Europe. The series of river discharge are in general discontinue in time due to problems in maintenance, impact of big floods, deinstallation of the instrument (Hanna et al., 2011; Crochemore et al., 2020). Our series are shown in Figure 1b, where the grey lines represent the period of continuous data for each station, listed based on the

latitude. We selected data from each station having a period of at least 10 years of continuous daily data, a time period similar to the expected life time of the sentinel mission (6 to 12 years; Drusch et al., 2012).

The selected timespan of the ground data is variable inside the working period of the hydrometric gauges and not necessarily co-exists with the Sentinel missions' lifetime. We selected those stations located in basins with area (Ab) greater than 100 km$^2$ and maximum observed discharge greater than 10 m$^3$/s. Indeed, smaller river discharges would generate flooded events short

in time and of negligible entity, difficult to map at the Sentinels' spatial resolution. The stations are grouped in four classes (Meybeck et al., 2006): B1 (Ab<500 km$^2$), B2 (500<Ab<5,000 km$^2$), B3 (5,000<Ab<15,000 km$^2$) and B4 (Ab>15,000 km$^2$), where Ab is the area of the basin in which the station was/is installed. The division is useful to analyse whether satellites are





able to detect floods in small basins like over larger basins. Figure 2 shows the distribution of the sites in terms of basin area, maximum and median value of river discharge. Respectively, the total number of sites in each group is 887 (B1), 746 (B2), 149 (B3) and 175 (B4). Only 17 % of the sites has a basin area larger than 5,000 with maximum discharge reaching 56,200 m$^3$/s (Pechora River at Oksino, Russia).

## 2.4 Working framework

In this work, we assume that it is always possible to map inundated areas from satellite images. This is quite confirmed by several examples in literature (Notti et al., 2018; Giordan et al., 2018; Musa et al., 2015; Schumann et al, 2015) and the operative Copernicus Emergency Management Service, in any case it represents a conservative assumption in favour of the usability of the systematic exploitation of the constellation. Therefore, the analysis concentrates on the chances that the satellite will intercept a flood.

In our analysis, we focused on floods caused by heavy and intense rains that provoked overflow of swollen water courses from their usual bed or basin. The overflowing is strictly dependent on the volume of water coming from the upstream and the cross section of the river. A volume of water flowing into a channel causes flooding when the cross section is not sufficient to contain it during the flow. Assuming that the cross-sections in the period of analysis remain unchanged (hence, no significant sediment loads affect the shape of the cross-section), we monitor the magnitude and the duration of the flood event through the analysis of the river discharge times series, or hydrographs. Figure 3a shows the hydrograph of the Loire River at the Mont Jean gauge station (belonging to B4 group with 110,000 km$^2$). Floods events tend to occur in correspondence of some extreme values of the hydrograph and stagnate for periods of some hours to several days depending on the conditions and the settings of the area. This temporal window can be suitable for the detection using satellite imagery when their frequency of acquisition is adequate. We assume here that a flood can occur when the river discharge overcomes a specific threshold. To identify the threshold for the river discharge, we assume that in 10 years flooding occurs for very high river discharge and that these high discharges are determined by setting thresholds equal or higher than the 95$^{th}$ percentile on the sample. For multi-peak event, if the peaks occur within four consecutive days, we consider it as a unique flood event. In the example at Figure 3a the selected river discharges cover the period from 1970 to 1980 and during this 10 years, the 95$^{th}$ percentile is overpassed 16 times, meaning that these 16 times the river flooded the floodplain around. To identify a reliable threshold, we created four scenarios using thresholds at 95$^{th}$, 97$^{th}$, 99$^{th}$ and 99.5$^{th}$ percentiles calculated in the same period of 10 years selected for the analysis and variable for each station.

To mimic the temporal sampling of Sentinel satellites observation, we assumed six different configurations based on the number of satellites in orbit: 12 days if only one satellite Sentinel-1 is in orbit, 6 days for two satellites A and B and 3 days if ascending and descending orbit are considered. For Sentinel-2, assuming the same configurations as before, we considered 10, 5 and 2 days (see Table 1). The simulation of the flood events captured by satellite observation was carried out by sampling the daily observed river discharge during the overpasses of the satellites according to the six configurations. From the river discharge time series sampled at the satellite temporal sampling, we counted how many times the different configurations



could have sampled the river discharge series in flooding situations (see Figure 3b-d). The comparison between the number of synthetic satellite events and the actual ground observed flood events provides the degree of reliability of the satellites to catch the inundation areas in Europe. We used the ratio between the number of events detected by Sentinel and the total number of events ground observed as performance index, F.

Along with the index F that defines the effectiveness of Sentinels to sample floods, the duration of the event helps us to understand its magnitude and its compatibility with the observation from space. Specifically, the duration of an event is determined according to the following steps: 1) select a flood event by calculating a local maximum in the temporal series, 2) separate the hydrograph between baseflow and event-flow, 3) calculate the magnitude (height) of the event by the distance between the minimum discharge event flow (step 2) and the peak value, 4) the duration is calculated as the width corresponding

at the half height. This approach is preferable with respect to consider the entire base width of the flood event because it does not take in account the baseflow of the hydrograph. Indeed, if we integrate the baseflow in the evaluation of a flood event the results can be biased from a large number of low values. Considering the width at half height we are more confident that the event can produce a flood inundation.

   Whereas Sentinel-1 is able to observe the Earth in all weather conditions, Sentinel-2 cannot penetrate clouds and because the

floods are usually caused by heavy and intense rainfall, the possibility to catch the high value of river discharge decreases. To take into account this phenomenon, we introduced in the analysis the cloud coverage concept. Specifically, we referred to the cloud dataset by Wilson and Jetz (2016, https://www.earthenv.org/cloud). The dataset provides the average monthly and annual percentage of clouds all over the world at 1 km of resolution. We extracted the values for each station and plotted the results in Figure 4. As expected, higher latitude show high mean percentage of clouds but Scandinavian countries and Island are less

affected on average by clouds with respect to Great Britain and Germany.

   We then multiplied the number of events potentially extracted by Sentinel-2 for the average annual percentage of free clouds (1 - probability of clouds) to simulate the rate of events that the optical sensor can detect.

## 3 Results and Discussion

   Here, the results of the synthetic analysis are described both in terms of index F and of duration of flood events. Same analysis

is shown for the three case studies in the actual conditions, to demonstrate the consistency and the robustness of the synthetic analysis. After, a paragraph of discussion lists some considerations along with the limits of the procedure.

### 3.1 Synthetic cases

   Based on the selected dataset, Figure 5 shows the box plot of the number of flood events extracted from the original observed time series based on the different thresholds for the four groups. The thresholds at 95th and 97th percentile include the higher

number of events that can be interpreted as unrealistic values: in fact, an average of 60 flood events extracted over the 95th percentile for B1 group are equivalent to 6 flood events per year on average that is not credible with respect to the numbers





published of the International Disaster Database (EM-DAT, 2019; as described in the introduction). The two thresholds at 99[th] and 99.5[th] percentile seem more reliable and plausible for an extraction of flood events that cause inundations, with values ranging on average from 20 to 10 and from 12 to 6 floods, respectively, in a period of 10 years.

Figure 6 shows the range of variability of the performance index F for each configuration and for each group of basins above the four different thresholds. With respect to the number of flood events observed by ground stations, no satellite configuration reach similar performance (F=100). As expected, an increased number of satellites provide an increased number of detected events: configurations S1A and S2A have low observed events, S1ABad and S2ABad are able to observe a greater number of flood events, whereas S1AB and S2AB are in the middle. Notwithstanding the revisit time of Sentinel-1 is greater than those

of Sentinel-2, the number of events potentially detected by the optical satellite is lower due to the average annual cloud coverage that affects the observations. As a result, all the configurations of Sentinel-2, intercept a reduced number of events with respect to those observed with Sentinel-1. On average, satellites are more suitable to observe floods over large basins. Median values of F for B4 are always greater than those of the other groups. Table 2 reports median values of the boxplots along with the median values of the analysis considering all the discharge dataset. The results for the entire dataset (all the

groups together) show that at the threshold of 99[th] percentile on average only the 20% of the flood events can be detected by one satellite of Sentinel-1 (18% if the threshold is 99.5[th]) and this percentage increases at the 58% if two satellites are in orbit and if we consider the ascending and the descending orbits (configuration S1Abad for both the thresholds 99[th] and 99.5[th]). However, this result is affected by the large number of basins owing to the group B1 and B2 with respect to the others of the groups B3 and B4. Indeed, if we look at the large basins higher percentages are obtained: the configuration S1ABad is able to

detect from 58% to 60% of the basins for B3 and from 63 to 67% for B4, based on the different thresholds. For Sentinel-2 the performances are slightly low because due to the presence of clouds. Only the 9% with the threshold of 99[th] percentile (10 % with the threshold at 99.5[th]) of the events can be observable by one satellite (S2A), whereas the best configuration (S2ABad) provide a coverage of 29% of the basins. Same conclusions of Sentinel-1 can be drawn for the different size of the basins for Sentinel-2: over large basins a greater number of flood events can be observed. In any case, in the configuration with more

observations, the 33% on the flood event can be observed and this is considerably lower than the one of Sentinel-1 (67%).
The different performances in the groups of basins is due to the duration of flood events. Plots in Figure 7 show the median values of the duration of the selected flood events extracted above the threshold at 99.5[th] percentile for every group.
In general, the duration of a flood event is below the three days (see the averaged distribution of durations in Table 3 for all the thresholds) for the groups of basins, and in particular the 50th percentile of the events has duration greater than the revisit

time related to the configuration S1ABad, therefore the possibility to observe flood events from this satellite becomes significant. Around the 25% of the events has duration greater than 6 days and around the 5% greater than 12 days. For the smaller thresholds (95[th], 97[th], 99[th] percentiles) results show that a lower number of events can be observed with the constellation of satellites and a few events with a single satellite.



## 3.2 Real cases

We tested the effectiveness of our framework in three study areas (two in B2 and one in B4) where important flood events occurred in the recent period:

-       Zaragoza station [latitude: 41.5950, longitude: -0.7702] along the Ebro River in Spain February – March 2015 (https://emergency.copernicus.eu/mapping/list-of-components/EMSR120),

-       Skelton station [latitude: 42.7919, longitude: -1.7886] along the Ouse River in England during the flood event of the
December 2015 occurred close to the York city (https://emergency.copernicus.eu/mapping/list-of-components/EMSR150),

-       Moncalieri station [latitude: 42.7919, longitude: -1.7886] along the Po River in Italy during the event of November-December 2016 (https://emergency.copernicus.eu/mapping/list-of-components/EMSR192),

We selected the dates of Sentinel-1 and Sentinel-2 image acquisition in Google Earth Engine cloud computing platform (Gorelick et al., 2017) according to the availability of ground-based observations. For the three study areas, Figure 8 shows
the river discharges observed at the gauged stations of Zaragoza (Figure 8a), Skelton (Figure 8b) and Moncalieri (Figure 8c) along with the acquisition dates of the available images from Sentinel-1 and Sentinel-2. Flood events mapped by the Copernicus Emergency Service are identified with the yellow background, while thresholds at 95th, 97th, 99th and 99.5th percentile are shown in dashed horizontal lines and they are calculated with respect to the selected.

For the case of Zaragoza, the hydrograph at the gauged station from the period 2015 – 2017, shows three big multi-peak flood
events, all extracted at the 95th percentile threshold (to be noted that for this station all the thresholds were calculated in the period 2004-2013). These high values of river discharge occurred in 2016 and 2017 provoked some ephemeral inundations in some meanders, but they did not affect the city and the settlements around. The flood event occurred in February-March 2015 is the only one extracted above the threshold of 99.5th that provoked a significant inundation, with flooded areas upstream and downstream the Zaragoza city. The event was well monitored by several images acquired by Sentinel-1A (the video in the
supporting material shows the evolution of the flood event from January to April 2015). The first moderate inundation occurred at the beginning of February 2015 perfectly observed by Sentinel-1A with two acquisitions on the 3rd and on the 4th February (see Figure 9a). After, a double peak flood occurred with substantial flooding until the beginning of March, and observed by Sentinel-1A on 5 March. From some days later, the water started to recede, leaving the ground saturated (compare Figure 9b and 9e), which favoured further flooding in the event at the end of March despite the fact that the flow values were not very
high (compare Figure 9f with 9a and 9c).

The basin at Zaragoza station belongs to the B4 group with a basin area around 40,000 km². F index calculated for the three year period is consistent with the violin plot of Figure 6 for all the four thresholds. In particular, around the 32% of flood events was correctly identified by Sentinel-1A, whereas a lower value of 10% of floods were observed by Sentinel-2 (specifically, a single image of 19 January 2017 is cloud free).

The basin at Skelton station belongs to the group B2 (basin area around 1,756 km²). F index calculated for Sentinel-1A is equal to 32%, 33%, 40% and 50% for the thresholds at 95th, 97th, 99th and 99.5th percentile (calculated for the period October 2003-





September 2013), quite in line with the synthetic analysis. Again, for Sentinel-2 the values are in the range 16 – 25% even if focusing on the December 2015 flood event, any peaks is observed by cloud free images. Only a cloud free image was acquired the 29 December 2015, showing large inundation over the York city. The flood event discharge was lower the threshold al 95th

percentile but being the flood event very large, the inundation lasted a long time before to recede.

Moncalieri station is not included in the dataset of the synthetic discharges selected in the analysis. The station is located along the upper part of the Po River with a basin area of 4,965 km$^2$. For this station, the four thresholds of percentiles were calculated for a period of 10 years from January 2010 to December 2019. Focusing on the period of three years of data, the flood events are a few and only the high values of discharge in November-December 2016 provoked inundation over the city of Moncalieri.

For this event, a couple of images acquired days later the peak value of 1,823 m$^3$/s were available. The inundation was visible only with SAR images being the Sentinel-2 images full of clouds.

### 3.3 Discussion

The above simulation can provide useful hints on the effectiveness on the use of Sentinel-1 and Sentinel-2 for systematic flood mapping. The real cases supported some assumptions in the analysis. Despite the procedure seems reliable for a good

evaluation of the effectiveness of Sentinels in the observation of flood events, we need to focus on several aspects to underline advantages and limitations.

First, the thresholds to extract flood peaks from river discharge (ranging from the 95th percentile to the 99.5th percentile) over a period of ten-year are arbitrary. These number of peaks above the thresholds can vary significantly, when analysing the data over a different period of time. Generally, the thresholds at 95th and 97th percentile extract an improbable number of flood

events (on average ranging from 62 and 25) whereas for the other two thresholds, 99th and 99.5th, the flood events are under a number of 20, meaning on average a couple of events per year (see Figure 5). This can be considered a good trade-off between the number of events and the possibility of inundation caused by the same events.

Second, during the period of 10 years, it is not necessarily each peak corresponds a flood. On one hand, the increase of discharge can be perfectly contained within the embankments without generate any floodplain inundation. On the other hand,

ordinary flood events can produce flooded areas if bridges are obstructed or if levees are broken. These cases are difficult to predict and/or simulate. However, the real cases showed in the study seem support the simplified procedure here proposed with a series of inundation caused by the biggest floods occurred in the three year period from 2015 to 2018.

Third, often the flooded areas can be observable from satellite for most of the duration of the recession limb of the hydrograph because terrain hit by the inundation remains wet for a long period. In this case, the stagnation time of the water in the

surrounding terrain may vary depending on several aspects: the type of soil, the topography (if the ground is flat or sloping may be different) and the volume of water overflowed. The case of Moncalieri station is a demonstration of the fact that after a flood event, even if the discharge is relatively low, it is still possible to observe inundations (in this case observable from Sentinel-2).



Four, we multiplied the number of events extracted by Sentinel-2 times the average annual value representing the percentage
of clouds for each site. Generally, a flood event caused by rainfall is accompanied by cloud cover and if we consider an average
annual value probably we overestimate the number of flood events that are possible to observe by satellite.

The analysis here presented cannot take in account all these factors endogenous. It is rather a rough assessment of the potential
of Sentinels on the evaluation of hydraulic risk in Europe. However, the three cases illustrated at Session 3.2 demonstrated
that the analysis is plausible and the Sentinels are able to catch the 30% and the 10% of the flood events.

## 4 Conclusions

As a tool to manage the emergency response after a flood inundation event, flood mapping helps to assess the extent of the
affected areas on a large scale. It is the base not only for the coordinating recovery activities, but also for preventing measures
of mitigation in case of upcoming events. In the last twenty years, a common practice is to map inundation events through the
Earth- observing satellites, especially with Synthetic Aperture Radar, SAR, or optical sensors. The Copernicus Emergency
Management Service, EMS, provides information for emergency response to a wide range of natural or man-made disasters.
Among them, the service covers floods. Indeed, one of the purposes of the Sentinels missions, particularly Sentinel-1 and
Sentinel-2, is to contribute to the Copernicus EMS and, more in general, to support disaster relief efforts, thanks to the short
revisit time and the rapid acquisition and delivery of the images.

To evaluate the effective capability of Sentinel-1 and Sentinel-2 to map floods in Europe, this paper carried out a synthetic
analysis measuring how many flood events the satellites can observe in their lifetime. To reach this target, we analysed the
daily river discharges that have been observed at 1,957 gauged sites along more than 1,300 rivers. We extracted the flood
events for every sites defining four thresholds of 95th, 97th, 99th and 99.5th percentile of each time series. Successively,
considering different configurations of revisit time for each satellite and assuming that each time the satellite overpasses the
river, it is able to acquire an imagery correctly, we estimated the percentage of flood events potentially observable by Sentinel-
1 and Sentinel-2.

From the analysis, the following conclusions can be drawn.

1 – On average, Sentinel-1 with a revisit time of 12 days is able to observe the 20% of flood events with a single satellite
whereas Sentinel-2, with 10 days of revisit time, is able to observe only the 10% with a single sensor. However, we need to
point out that for Sentinel-2 this percentage can be also overestimated because the clouds are generally intense during flood
events.

2 – More satellites increase the chances of observing a flood event. If two satellites constellation is orbiting and both ascending
and descending orbits are considered, the percentages of flood event potentially observable increase up to 58% for Sentinel-1
and 28% for Sentinel-2.

3 – The ability of satellites to observe a flood event in a site changes depending on the size of the catchment area subtended
by the section. In the configuration of more orbiting satellites, and higher threshold (99.5th percentile) Sentinel-1 is able to



detect the 50% of flood events at sites with basin area smaller than 500 km$^2$, and the 67% of flood events at sites with basin area greater than 15,000 km$^2$. In the similar configuration, the percentages of flood events observable by Sentinel-2 vary from 23% for the smaller basins to 33% for the bigger basins.

4 – The advantage of having stagnant water in the floodplain areas after the flood peak makes it possible to observe the extent of the flooding even after several days. However, when analysing the average duration time of a flood event, in most cases this value is around 2.5 - 3.5. Therefore, if the satellite orbit were programmed to have an average revisit time of 2 days, almost all events could be mapped.

**Video supplement:** the video is available in the supplementary material.

**Code availability:** the code is in Matlab and it will be available after the publication of the paper.

**Data availability:** the river discharge data are available from the Global Runoff Data Centre, (https://www.bafg.de/GRDC/EN/01_GRDC/grdc_node.html). The Sentinel-1 and Sentinel-2 data from the real case were taken from the Google Earth Engine platform (https://code.earthengine.google.com/), whereas the information about the flood events are provided by the Emergency Response and Coordination Centre (ERCC) at DG ECHO, (http://emergency.copernicus.eu).

**Author contribution:** A.M. designed the experiments, A.T., developed the model code and performed the simulations. S.C. provided the in situ data of river discharge and contributed to the model code. A.T. prepared the manusctipt with contributions from all co-authors.

**Acknowledgments**: A.T., S.C., A.M. acknowledge the support of European Space Agency through the Project COMMONs (SEOM SY 4Sci Synergy N. contract ESRIN/1-7831/14/I-NB).

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

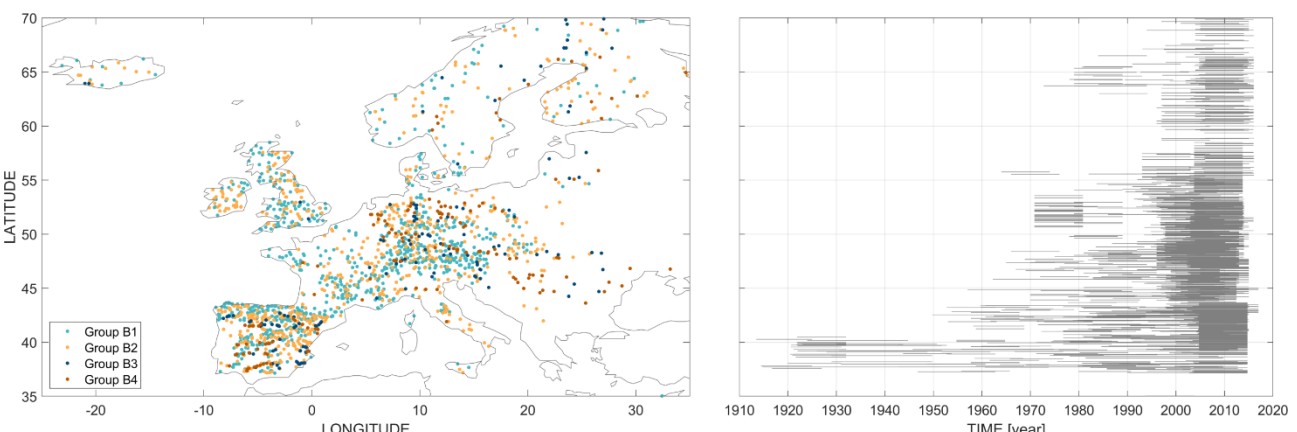

**Figure 1: a) location of the selected gauged stations where the river discharge is recorded and b) operating period for each ground station from 1913 to 2016 listed based on the latitude**

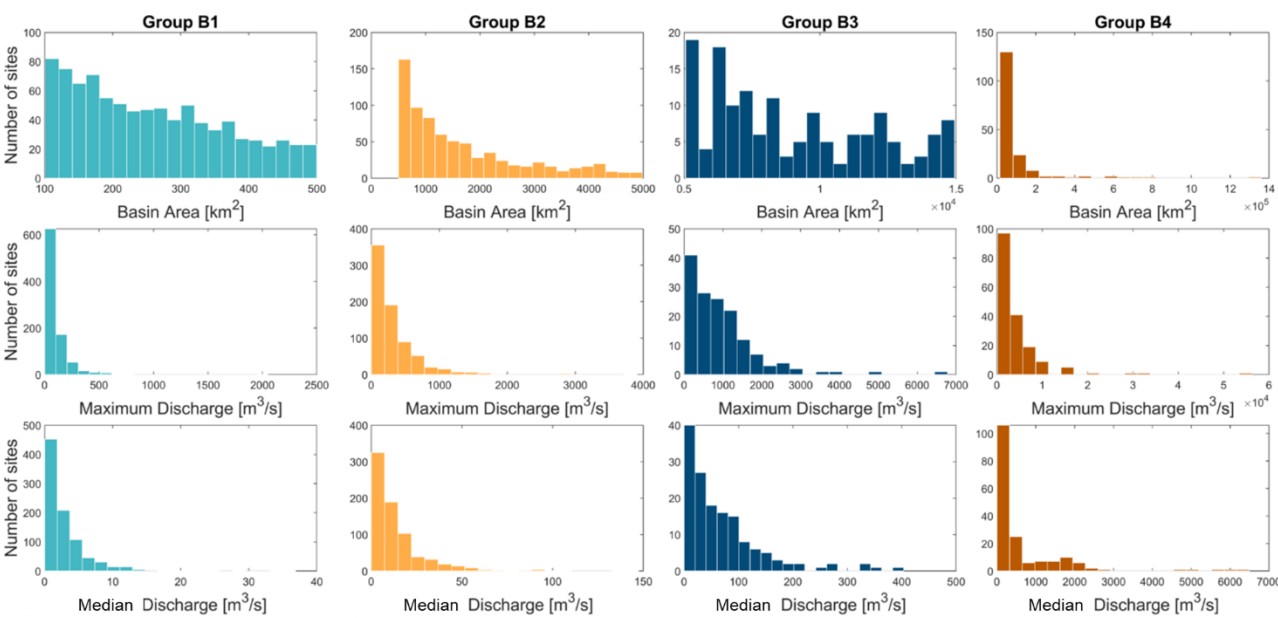

**Figure 2: Histograms of the basin area [km²] (top), maximum river discharge [m³/s] (middle) and median discharge [m³/s] (bottom) of the ground European dataset. The 1957 stations are divided in four groups based on the basin area, Ab: B1 (Ab<500 km²), B2 (500< Ab<5,000 km²), B3 (5,000< Ab<15,000 km²) and B4 (Ab>15,000 km²).**






**Figure 3: Example of events selection by ground observed daily discharge (a) and assuming satellite revisit time of 3 (b), 6 (c) and 12 days (d) for the Loire River at Mont Jean gauged site.**




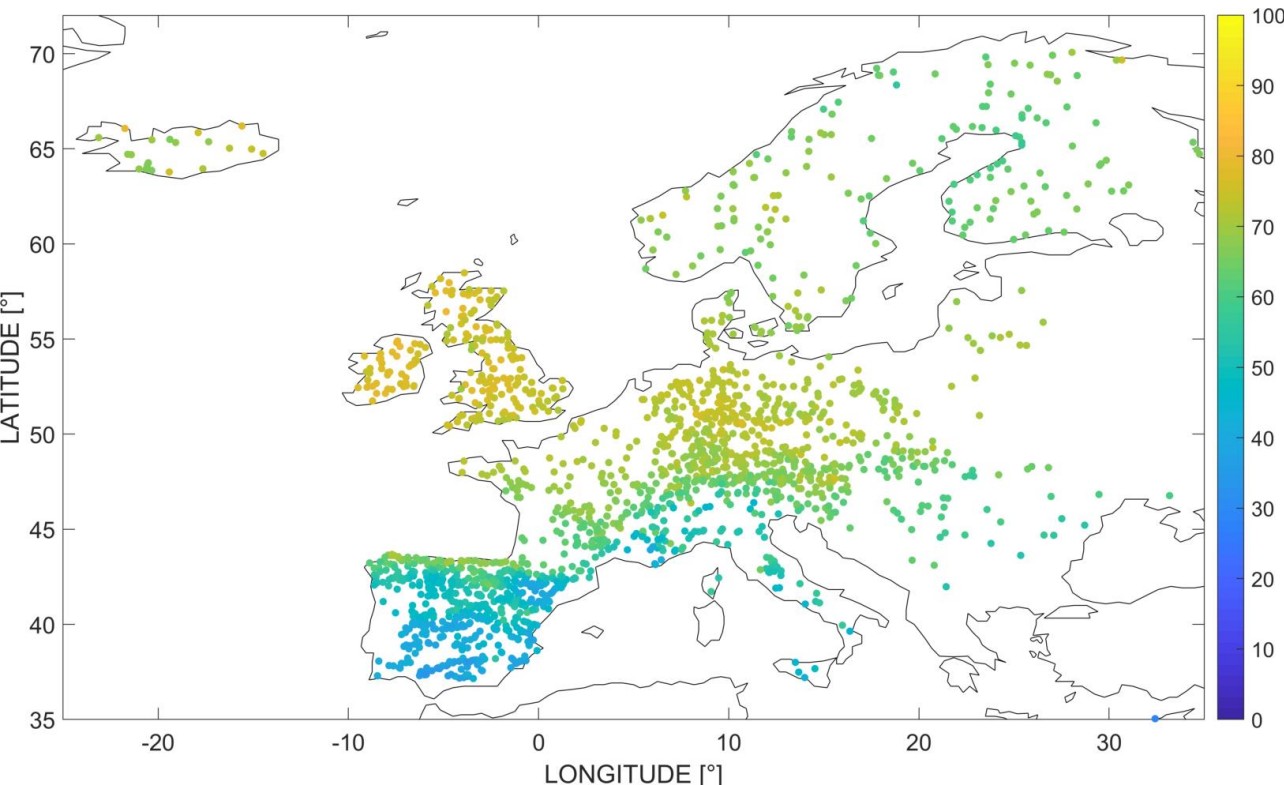

**Figure 4: Annual percentage of clouds for the 1957 selected stations.**

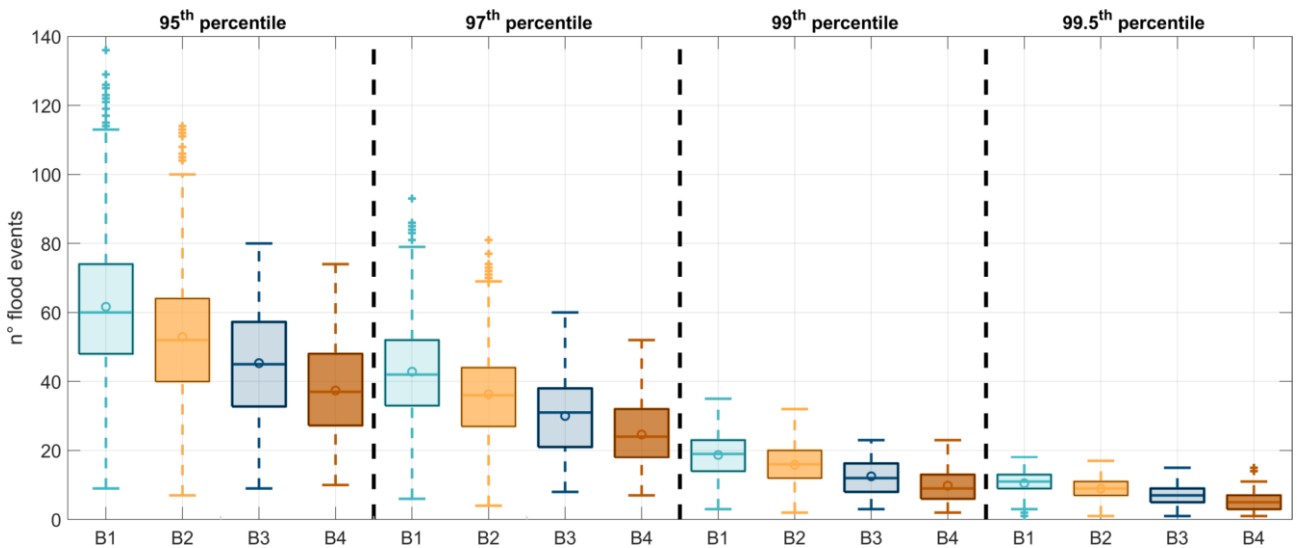

**Figure 5: Box plot of the number of flood events extracted by selecting four different thresholds (95th, 97th, 99th, 99.5th percentile) over the four different groups of basins (B1, B2, B3, B4).**



**Figure 6: Violin plot of the percentage of events detected by satellite observations with respect to the total events observed at ground sites. The six scenarios are explained on Table 1.**


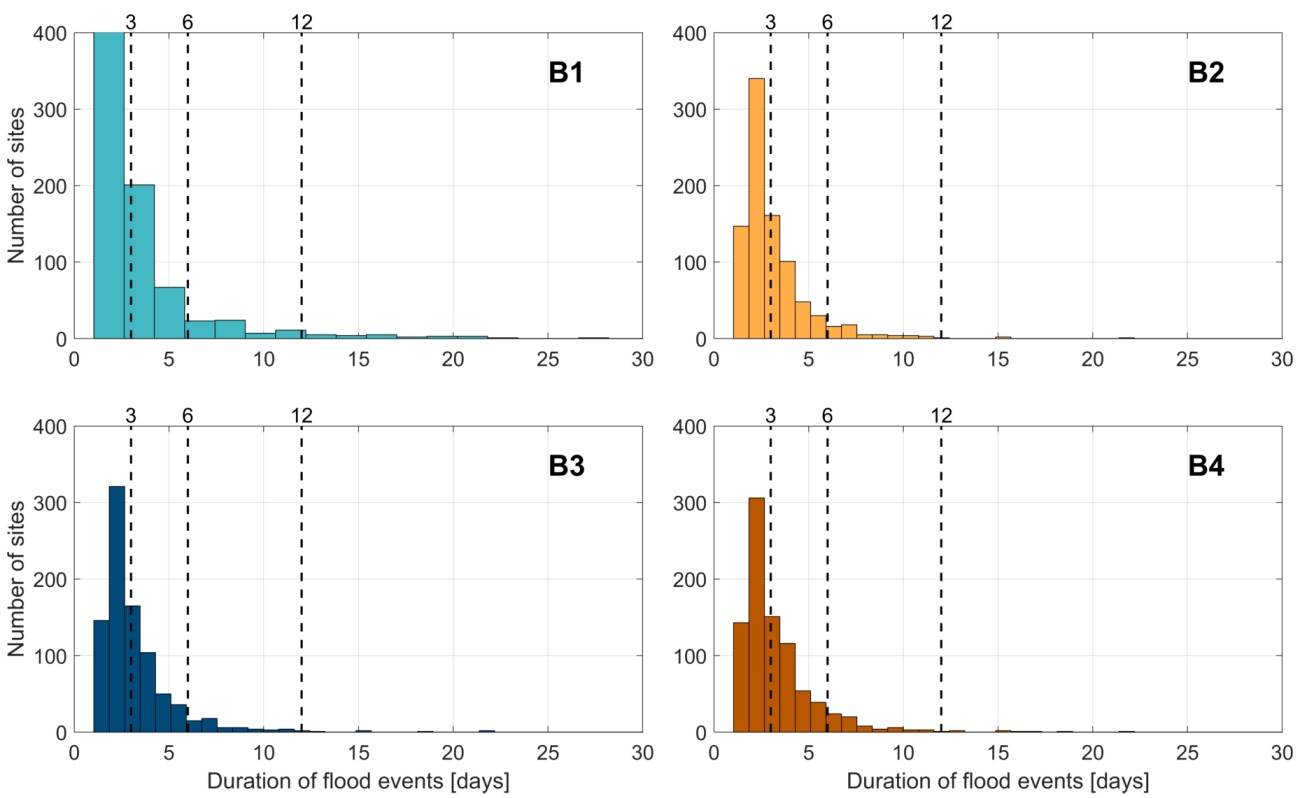


**Figure 7: Median values of duration of the flood events extracted by a threshold of 99.5th percentile, for each group of basins (B1, B2, B3 and B4). The dashed lines represent the revisit time relate to the three configurations of Sentinel-1: 3, 6 and 12 days.**



**Figure 8: river discharges observed at the gauged stations of Zaragoza (a), Skelton (b) and Moncalieri (c) along with the acquisition dates of the available images from Sentinel-1 (circles) and Sentinel-2 (plus). The period of flood event selected by Copernicus Emergency Service is identified with the yellow background, while dashed horizontal lines identify the thresholds at 95th, 97th, 99th and 99.5th percentile.**

**Figure 9: Flood event occurred in the period February - March 2015 at Zaragoza (g). The letters in the hydrograph correspond to the SAR images on the top (©Google Earth Engine).**



**Table 1: Configurations used in the study for different revisit time based on the combinations of satellites**

| Name | Configurations | Revisit time (days) |
|---|---|---|
| S1A | Sentinel-1 one satellite | 12 |
| S1AB | Sentinel-1 two satellites | 6 |
| S1ABad | Sentinel-1 two satellites ascending and descending | 3 |
| S2A | Sentinel-2 one satellite | 10 |
| S2AB | Sentinel-2 two satellites | 5 |
| S2ABad | Sentinel-2 two satellites ascending and descending | 2 |


**Table 2: F median value for the satellite configurations and for the four groups of sites (values are in terms of percentage, %).**

| | 95th percentile | | | | 97th percentile | | | |
|---|---|---|---|---|---|---|---|---|
| | B1 | B2 | B3 | B4 | B1 | B2 | B3 | B4 |
| FS1A | 0.19 | 0.20 | 0.23 | 0.26 | 0.18 | 0.20 | 0.22 | 0.27 |
| FS1AB | 0.33 | 0.35 | 0.37 | 0.42 | 0.32 | 0.35 | 0.37 | 0.44 |
| FS1ABad | 0.54 | 0.57 | 0.58 | 0.63 | 0.53 | 0.57 | 0.59 | 0.67 |
| FS2A | 0.08 | 0.10 | 0.11 | 0.13 | 0.07 | 0.09 | 0.11 | 0.13 |
| FS2AB | 0.13 | 0.16 | 0.17 | 0.20 | 0.13 | 0.16 | 0.18 | 0.21 |
| FS2ABad | 0.24 | 0.27 | 0.31 | 0.29 | 0.24 | 0.27 | 0.29 | 0.30 |
| | 99th percentile | | | | 99.5th percentile | | | |
| | B1 | B2 | B3 | B4 | B1 | B2 | B3 | B4 |
| FS1A | 0.15 | 0.18 | 0.21 | 0.27 | 0.13 | 0.14 | 0.20 | 0.25 |
| FS1AB | 0.29 | 0.33 | 0.38 | 0.44 | 0.27 | 0.29 | 0.33 | 0.43 |
| FS1ABad | 0.52 | 0.55 | 0.59 | 0.67 | 0.50 | 0.55 | 0.60 | 0.67 |
| FS2A | 0.06 | 0.08 | 0.11 | 0.14 | 0.07 | 0.08 | 0.13 | 0.13 |
| FS2AB | 0.12 | 0.14 | 0.17 | 0.22 | 0.10 | 0.13 | 0.18 | 0.22 |
| FS2ABad | 0.24 | 0.27 | 0.30 | 0.33 | 0.23 | 0.27 | 0.31 | 0.33 |

**Table 3: Average duration in days of a flood event belonging to the different groups and extracted above the different thresholds.**

| Group | 95th | 97th | 99th | 99.5th |
|---|---|---|---|---|
| B1 | 2.66 | 2.82 | 3.24 | 3.48 |
| B2 | 2.52 | 2.66 | 2.97 | 3.14 |
| B3 | 2.54 | 2.68 | 3.02 | 3.26 |
| B4 | 2.60 | 2.73 | 3.08 | 3.36 |