# Peer review of "Effectiveness of Sentinel-1 and Sentinel-2 for Flood Detection Assessment in Europe"

_Natural Hazards and Earth System Sciences, 2022_

## Referee Comment (RC2)

[referee-annotated manuscript omitted]

---

## Author Comment (AC1)

We would like to thank the Associate Editor and the reviewers for the valuable feedback to the preprint Discussion.

Please note:

Responses to reviewers were marked with blu color and changes to the text with red color.

**REFEREE#1**

The authors attempt to assess which percentage of flood events can theoretically be observed by the satellites Sentinel-1 and -2. They do so by a rather coarse synthetic study, with several optimistic assumptions, but I very much appreciate the research question and the clarity with which the authors approach the topic. The paper is very well written, key messages are listed in the conclusions, figures are good quality. The authors discuss most of their assumptions, there are just a few points on which I request further clarification:

- We would like to thank the reviewer for the comments raised and for the stimulating review. We were very pleased to read it and we did our best to improve the discussion based on the points mentioned by the reviewer.

1. A lot of SAR data is unfortunately commercial, Sentinel-1 being a notable exception

   - Indeed, and the same for optical sensors. We will modify the sentence in the introduction accordingly:
     "The availability of these two instruments, optical and SAR, in the satellite platform assurances a global coverage and, in some of the cases, a cost-free source of data"

2. It is absolutely fine as a "working framework" to assume that it is always possible to map inundated areas from satellite images, however to state that this assumption is "quite confirmed by several examples in literature" is misleading. There are still big issues with satellite-based flood mapping! A general discussion on the limitations can for example be found in Schumann 2021: https://doi.org/10.1016/B978-0-12-819412-6.00014-6 and a more specific evaluation on a well-documented flood, mapped by the Copernicus EMS and other scientific products with a good ground-truth reference, revealed that rapid mapping products can be of very poor quality (Table 3): https://doi.org/10.3390/rs13112042 This is not only a problem of the classification algorithms, but indeed also of the image quality / observability. In sensor design there is a trade-off between image quality and coverage, so when investigating the potential coverage of specific satellites, we have to deal with the true quality of the sensor. Another key difference between optical and SAR data is the viewing geometry. Operational SAR sensors are side-looking, which causes additional issues in urban areas, radar shadows, layover effects etc. Therefore, it is in theory able to detect water below vegetation, however that depends on wavelength and is usually not part of operational flood detection algorithms. Ignoring flooded vegetation can obscure the true land-water boundary, and most impacts occur in urban areas. Products like EMSR detect almost exclusively open water, and even that is not always convincing! There should be a paragraph in the paper on these limitations, to avoid the impression that the 58% potentially observable flood events by Sentinel-1 actually translate to 58% of flood events being mapped in sufficient quality, I assume it is only a small fraction of that.

   - We agree with the reviewer and we specified this as a "conservative assumption". In any case, we will modify the sentence as follows:

"In this work, we assume that it is always possible to map inundated areas from satellite images as shown by several examples in literature (Notti et al., 2018; Giordan et al., 2018; Musa et al., 2015; Schumann et al, 2015) and the operative Copernicus Emergency Management Service. However, we are aware that this is a strong assumption (see paragraph 3.3 Discussion for details) and it can be acceptable for a synthetic study in favour of the usability of the systematic exploitation of the constellation."

and we will add all these points the reviewer raised in the Discussion section as in the sequel:

"Fifth, the initial assumption to consider always possible to map inundated areas from satellite images is an optimistic and unrealistic hypothesis for several reasons (Schumann 2021): i) the presence of frequent, persistent and large-scale cloud cover is the most severe aspect to consider for optical sensors especially in case of flash floods; ii) the vegetation is still an arduous problem for both optical satellite sensors (DeVries et al., 2017) and SAR, because of the side-looking nature and the diffusive and volume scattering caused by vegetation (Schumann and Moller 2015); iii) the urban areas, present important challenges above all for SAR due to inadequate spatial resolution of the sensor and man-made features, which cause a lot of signal distortion (Chini et al., 2012; Giustarini et al., 2013); iv) the quality of the images, especially of the rapid mapping products can be very poor (Brill et al., 2021). Therefore, the percentages extracted from this analysis can be seen as an optimistic view of potentially observable flood events, but which should be translated, into reality, in much lower percentage values, if we are to address the challenges mentioned above and demand sufficient quality."

3. The authors made an assumption on cloud coverage based on a dataset by Wilson and Jetz 2016. They do not take into account that flood events are typically triggered by rain, which requires clouds, and therefore should expect a correlation between the presence of clouds and a flood event. The assumption of the authors is therefore optimistic, which is ok as long as it is clearly stated. I would find it interesting to actually check how often and how long floods are accompanied by clouds, depending on the geolocation/climate, but I understand that this was not aim of the study to do so. My feeling is that there could be quite significant spatial differences on the percentage of floods that optical sensors may detect (while for SAR it should be the same percentage in all places). Detectability on SAR images probably depends more on topography or built-up density. As the two percentages are your primary results, please briefly discuss this point and whether you think it is useful/possible to put a number on that spatial variability.

- We recognize that the dataset by Wilson and Jetz (2016) is not comprehensive and exhaustive for this analysis, but it already provides a spatial variability of cloud coverage (see Figure 4). In terms of correlation between the presence of clouds and a flood event, we already considered this limitation in the fourth point described in Session 3.3 Discussion. In any case, we added some comments related to the point suggested by the reviewer and now it reads:
"Four, we multiplied the number of events extracted by Sentinel-2 times the average annual value representing the percentage of clouds for each site. Generally, a flood event caused by rainfall is accompanied by cloud cover and if we consider an average annual value probably we overestimate the number of flood events that are possible to observe by satellite. In fact, clouds are concentrated during flood events rather than for low flows, and this exacerbates the difficulties of mapping the inundation with an optical sensor. However, the advantage is that in many cases the

water remains in the flooded areas for a while favouring mapping even several days later: unfortunately, in those cases the mapping of the maximum extent of the flood is compromised.

- The request of the reviewer on how often and how long floods are accompanied by clouds, depending on the geolocation/climate is not easy. We regret that we cannot satisfy the reviewer, but as it is not the purpose of this study, we prefer not to calculate it.

4. Another debatable assumption (which the authors do mention) is the definition of a flood event by placing a percentile threshold on a 10-year discharge (!) observation time series. Whether a flood occurs or not is of course dependent on the protective measures, which drastically vary in their design level, up to > 1/10000 years in the Netherlands https://www.deltares.nl/app/uploads/2014/12/kind2014_JFRM1.pdf In rural areas, if there is nothing except agricultural crops to protect, actual flood defense might be much lower, but this will probably not be at the location of the measurement station? There are footprints of real flood events, e.g. from the Dartmouth flood observatory that could potentially be used for such a purpose. See Figure 6 in Lüdtke et al. 2019: https://doi.org/10.1029/2019WR026213 I am not entirely sure why the authors have not used validated flood locations, but I do find the synthetic approach also very interesting. Maybe you can make this more clear?

- Honestly, the definition of thresholds is one of the most critical points in this analysis. A period of 10 years was selected because it is compatible with the lifetime of the satellites and can be considered a good compromise between continuous data availability and the occurrence of a significant event. In fact, it is desirable that in places where the measuring instrument is present, which typically coincides with the most critical areas (urban areas, confluences,...), protective measures have been put in place to prevent flooding, but recent events in Europe have shown that extreme events are abundantly higher than a few years ago and that the protective measures taken previously are no longer sufficient to protect against flooding (e.g., the extreme flood event occurred last July in Germany, Luxembourg…). This point was partially covered in the discussion, but we will integrate the discussion as follows:

  "Second, during the period of 10 years, it is not necessarily each peak corresponds a flood. On one hand, the increase of discharge can be perfectly contained within the embankments without generate any floodplain inundation. On the other hand, ordinary flood events can produce flooded areas if bridges are obstructed or if levees are broken. These cases are difficult to predict and/or simulate. In addition, it is desirable that in places where the measuring instrument is present, which typically coincide with the most critical areas (urban areas, confluences, ...), protective measures have been put in place to prevent flooding. However, recent events in Europe have shown that extreme events are abundantly higher than they were a few years ago, and that the protection measures taken previously are no longer sufficient to protect against flooding. In any case, the real cases showed in the study seem support the simplified procedure here proposed with a series of inundation caused by the biggest floods occurred in the three years period from 2015 to 2018."

I am happy to recommend the article for publication, if the abovementioned discussion points are addressed.

**REFEREE#2**

Tarpanelli et al., address an important question in the submitted manuscript, namely the suitability of the Sentinel 1/2 Satellites for flood inundation mapping in Europe. Through synthetic assessments based on discharge values to detect flood events, the authors simulate the satellite coverages and calculate the probability of capturing inundation events through the Sentinel-1 Synthetic Aperture Radar sensor and the Sentinel-2 optical sensor. The study also tests the findings from the synthetic study for three real world flood events and find most of their conclusions supported through this analysis. In principle the paper is well written, easy to follow, and of interest to the larger flood mapping community in Europe. I only have some minor comments as summarized below, which I believe will help improve the quality of the manuscript.

- We would like to thank the reviewer for the comments and suggestions. We will implement them in the revised version of the manuscript.

1. Referencing: the introduction cites papers from over a decade ago to establish current flood impacts and scientific advances. Given the rapidly evolving body of literature in this topic, I think this should be improved as newer publications are sometimes tackling the newer challenges in the field which arose out of the rise of big data and machine learning. I have provided some reference suggestions, but the authors are welcome to seek out some more. I also found some references cited wrongly – e.g. Clement et al. 2018 is cited for the extraction of inundation water levels which they did not actually do in the paper. The authors should check such oversights in the referencing throughout and correct these before publication.

    - We really thank the reviewer for the careful read of the paper and for the precious suggestions of the references. We updated the literature with new studies and we will remove the references of Clement et al. (2018).

2. Recent relevant developments: For a study looking to assess the suitability of the Sentinels for flood monitoring in Europe, the paper misses two very important and relevant new developments. These are namely, the launch of the Global Flood Monitoring Service from the Copernicus Emergency Management Services and the failure of Sentinel-1b. While it might not be possible to account for the latter in the analysis at this time without needing to reproduce the figures, I think it is still relevant to acknowledge this issue either in the introduction or in the discussion/conclusions and the potential impact this has on the conclusions of this study.

    - Actually, the option to consider a single sensor in orbit was already considered in the analysis. Therefore, it is sufficient to update the text that will be modified as follows. At the end of the Session 2.1 - Sentinel-1 SAR imagery and their current use in flood detection/mapping- we will add:
    "However, Sentinel-1B launched in April 2016 malfunctioned in December 2021 due to power issue, with consequence loss of data transmission.

Concerning the Global Flood Monitoring product, we will add in the introduction as specified as follows:

"In 2021, a new operational, near real-time global flood monitoring (GFM) was integrated into GloFAS. The new GFM analyses all the incoming Sentinel-1 images through 3 independently state-of-the-art satellite flood mapping algorithms (HASARD, ALGORITHM2, and ALGORITHM3) and provides a high timeliness and quality product based on the ensemble flood mapping."

3. Relationship between discharge and inundation: The authors assume that the discharge peaks represent the inundation peaks as well, this is not true in most cases due to the lag between the channel and floodplain peaks, as well as the highly non-linear relationship between discharge and inundation. Again, I do not think there is any need to alter the analysis, however, it would be nice to have the authors acknowledge this point while stating their assumptions and then assess the potential impact this may have on their conclusions in the discussion section.

- We agree that the time of peak discharge does not correspond to the time of peak flooding. In general, the two are shifted by a time lag that can vary, and is difficult to predict. Obviously, this aspect was not considered because it would be very difficult to quantify a lag that is adaptable to all streams. We thought that to reconstruct floods synthetically, the only way forward was to analyse high flows and the duration of flood events. However, we take the reviewer's suggestion and clarify this in the text by including this sentence in the discussion:

"In doing so, the time of peak flow is considered to coincide with the time of peak flood. In reality, the two moments are shifted by a lag time that varies with the properties of the basin and channel. We can reasonably accept this approximation considering that the phenomenon is not immediate, but that the maximum flood generally occurs sometime after the peak flow and still remains for a few hours (even days). The satellite may pass over the river during this time and be able to capture the flooded area. In this case, instead of associating the inundation with a probably lower river discharge, we prefer to associate it with the event that causes it."

Technical corrections are very few and included in the reviewed PDF file. On incorporating these minor comments, I think this article would form a valuable addition to the published literature in this direction. I look forward to seeing the final version online and thank the authors for their time and efforts.

- Thank you very much for this careful and comprehensive reading of the paper. In the following the specific comments are reported and discussed.

**Specific comments from the pdf**

C1: Line 29: The reference is 10 years old, is it maybe possible to use a more recent one? How about:

IFRC. (2020). World Disasters Report 2020: Come Heat or High Water. In World Disaster Report 2020. https://media.ifrc.org/ifrc/world-disaster-report-2020%0ACover

- R1: thank you for the suggestion. The citation will be replaced in the revised manuscript.

C2: Line31: The reference is of 2019, I am not sure this includes information from 2020-21.

- The International Disaster Database is an updated database consultable online (EM-DAT | The international disasters database (emdat.be)
  The citation in the main text represents the citation of the study in which the database is described.

C3: Line 41: reference required to support the statemen

- The references of Moramarco et al., 2014; Massari et al., 2015; Schumann et al., 2011 have been added.
- Massari C., Tarpanelli A., Moramarco T. (2015) A fast simplified model for predicting river flood inundation probabilities conditioned on flood extent data. Hydrological processes,29(10), 2275-2289. http://dx.doi.org/10.1002/hyp.10367
- Moramarco, T., Barbetta, S., Pandolfo, C., Tarpanelli, A., Berni, N., Morbidelli, R. (2014). The spillway collapse of the Montedoglio dam on the Tiber River (central Italy): data collection and event analysis. J Hydrol Eng, 19(6), 1264–1270, doi:10.1061/(ASCE)HE.1943-5584.0000890
- Schumann, G.J.P., Neal, J.C., Mason, D.C. and Bates, P.D.: The accuracy of sequential aerial photography and SAR data for observing urban flood dynamics, a case study of the UK summer 2007 floods, Remote Sens. Environ., 115(10), 2536 -2546, doi:10.1016/j.rse.2011.04.039, 2011.

C4: Line 47-48: maybe separate the references into the different scales mentioned? Also Bates and De Roo proposed the LISFLOOD-FP model and that too not in its currently popular form which is from Bates 2010. perhaps its worth using more specific references.

- We follow the suggestion of the reviewer, rephrase the sentence and add more recent studies:

"Flood inundation modelling including empirical, hydrodynamic and simple conceptual models can provide a valuable tool for delineating the flooded area and many examples can be found in literature (see Teng et al., 2017; Mudashiru et al., 2021; for a review). Often the modelling is carried out before the flood event occurs in order to collect inundation scenarios and identify the edge of the potential flooded areas."

C5: Line 60: Clement et al do not extract inundation levels.

- Clement et al. 2018 was removed.

C6: Line 78-81: I think two very relevant new developments deserve mention here:

1. The recently operational Global Flood Monitoring Service product by the CEMS which is filling exactly this gap

2. The failure of Sentinel-1b which has severely impacted revisit estimates for the moment until the next Sentinels are operational in orbit

- The following sentence will be added:

"In 2021, a new operational, near real-time global flood monitoring (GFM) was integrated into GloFAS. The new GFM analyses all the incoming Sentinel-1 images through 3 independently state-of-the-art satellite flood mapping algorithms (HASARD, ALGORITHM2, and ALGORITHM3) and provides a high timeliness and quality product based on the ensemble flood mapping."

"Moreover, Sentinel-1B launched in April 2016 malfunctioned in December 2021 due to power issue, with consequence loss of data transmission."

C7: Line 120: *"Oberstandler et al. (1997) introduced the visual interpretation approach, stating that it is able to provide more accurate results than an automatic procedure and many studies were based on that."* However, the recent rise of Big data in EO makes this nearly impossible no? Maybe relevant to acknowledge this?

- The following sentence will be added:

"However, such methods, although robust and reliable, have mostly been developed for a few images and demonstrated only at local level. With the proliferation of spatial data (Schumann & Domeneghetti, 2016) there is a great emphasis on globally scalable algorithms powered by artificial intelligence and machine learning and big data analytics. In such prospective, a rigorous validation at global scale of satellite products is fundamental and strongly recommended (Schumann, 2019)."

C8: Line 155: *"negligible entity"* ??

- We will change in "negligible peak values"

C9: Line 169: *"The overflowing is strictly dependent on the volume of water coming from the upstream and the cross 170 section of the river."* lateral flows? roughness?

- We will add also lateral flows and roughness

C10: Line 176: *"This temporal window can be suitable for the detection using satellite imagery when their frequency of acquisition is adequate."* or by luck if the floods correspond with the overpass?

- Our statement means that if we can consider a long period we can have more chances to map a flood event by an overpassing of a satellite. If the frequency of the satellite is dense (2 passages per day) we are quite sure to map the flood event.

C11: Line 191: *"From the river discharge time series sampled at the satellite temporal sampling, we counted how many times the different configurations could have sampled the river discharge series in flooding situations (see Figure 3b-d)."* Yes, but the highly non-linear relationship between discharge and inundation implies that the time of peak discharge and the time of peak inundation are lagged, and this lag varies based on catchment and channel properties. How is this accounted for in the assumptions?

- We agree that the time of peak discharge does not correspond to the time of peak flooding. In general, the two are shifted by a time frame that can vary, and is difficult to predict. Obviously, this aspect was not considered because it would be very

difficult to quantify a lag that is adaptable to all streams. We thought that to reconstruct floods synthetically, the only way forward was to analyse high flows and the duration of flood events. However, we take the reviewer's suggestion and clarify this in the text by including this sentence in the discussion:

"In doing so, the time of peak flow is considered to coincide with the time of peak flood. In reality, the two moments are shifted by a lag time that varies with the properties of the basin and channel. We can reasonably accept this approximation considering that the phenomenon is not immediate, but that the maximum flood generally occurs sometime after the peak flow and still remains for a few hours (even days). The satellite may pass over the river during this time and be able to capture the flooded area. In this case, instead of associating the inundation with a probably lower river discharge, we prefer to associate it with the event that causes it."

C12: Line 194: formula?

- We believe that it is not necessary to add a simple formula already described in the main text as the ratio between the number of events detected by Sentinel and the total number of events ground observed.